# Modeling influenza transmission dynamics with media coverage data of the 2009 H1N1 outbreak in Korea

**Yunhwan Kim**[1], **Ana Vivas Barber**[2], **Sunmi Lee**[3,4]*

1 Division of Media Communication, Hankuk University of Foreign Studies, Seoul, Korea, 2 Department of Mathematics, Norfolk State University, Norfolk, Virginia, United States of America, 3 Department of Applied Mathematics, Kyung Hee University, Yongin, Korea, 4 Institute of Natural Sciences, Kyung Hee University, Yongin, Korea

* sunmilee@khu.ac.kr

**Data Availability Statement:** All relevant data is within the paper and its Supporting Information files.

**Funding:** This work was supported by a National Research Foundation of Korea (NRF) grant funded

## Abstract

Recurrent outbreaks of the influenza virus continue to pose a serious health threat all over the world. The role of mass media becomes increasingly important in modeling infectious disease transmission dynamics since it can provide public health information that influences risk perception and health behaviors. Motivated by the recent 2009 H1N1 influenza pandemic outbreak in South Korea, a mathematical model has been developed. In this work, a previous influenza transmission model is modified by incorporating two distinct media effect terms in the transmission rate function; (1) a theory-based media effect term is defined as a function of the number of infected people and its rage of change and (2) a data-based media effect term employs the real-world media coverage data during the same period of the 2009 influenza outbreak. The transmission rate and the media parameters are estimated through the least-squares fitting of the influenza model with two media effect terms to the 2009 H1N1 cumulative number of confirmed cases. The impacts of media effect terms are investigated in terms of incidence and cumulative incidence. Our results highlight that the theory-based and data-based media effect terms have almost the same influence on the influenza dynamics under the parameters obtained in this study. Numerical simulations suggest that the media can have a positive influence on influenza dynamics; more media coverage leads to a reduced peak size and final epidemic size of influenza.

## 1 Introduction

Biological and social aspects are critical factors in the transmission dynamics of human infectious diseases. The biological characteristics of pathogens play a critical role in how easily an individual is influenced by the pathogen, how long the latent period might be, and how severe the disease might be. The individual-to-individual transmission of diseases is greatly dependent on social aspects of human populations. The way that interactions among individuals are structured can determine whether a disease spreads throughout a population or dies out. Thus intervention strategies are devised, focusing on social aspects such as school-closing or social

by the Korean government (MSIP) (NRF-2018R1A2B6007668).

**Competing interests:** The authors have declared that no competing interests exist.

distancing, as well as biological aspects such as vaccination or anti-viral treatment. For instance, whether and how to make people participate in a vaccination program can be considered a social problem because psychological factors such as vaccine scares can affect the vaccination behavior of individuals [1]. Those two aspects affect each other, and the mode of this mutual influence determines the mode of disease spread [2].

There has been much effort towards incorporating such social factors into mathematical models [3]. They facilitate or hinder human behaviors, which are related to epidemics based on the assumption that people may adopt behaviors that reduce the risk of disease if they know about it. Since the mass media is both the easiest way of disseminating information to a large number of people and the major information source regarding disease outbreak, mass media is one of the factors that has drawn the most attention. With many headings such as mass media coverage [4, 5], awareness and psychological impact [6–8], the role of mass media on the spread of infectious disease has accounted for a significant amount of literature about mathematical modeling of social aspects in epidemiology.

Previous studies about the effects of mass media on the spread of infectious diseases can be categorized into three kinds according to the way that the media effect is represented in mathematical equations. There are studies that divide classes in the SIR-type model by the response to mass media. Funk et al. [6] separated each of the classes in the SIR model into either being aware or unaware of the disease. Individuals can be aware of the disease in any class of the SIR model, and individuals' awareness affects the infection rate, recovery rate, and rate of immunity loss. By conducting mean-field analysis and numerical simulations, they found that spreading awareness of an endemic disease, which is identical to the media effect, can make it impossible for the disease to settle in a population under certain conditions. They have also shown that the media can decelerate the spread of the disease. Kiss et al. [9] reached similar conclusions with a modified SIRS model. They divided the classes into responsive and non-responsive, and they considered three scenarios. The first is where information about a disease is transmitted by contact among individuals, the second one is where information is spread population-wide, and the third one is where the value of the information decays over time. They showed that population-wide information transmission, which is equivalent to the media effect, is less effective in resisting disease spread than contact-based information transmission. Misra et al. [7] utilized a simpler model, which divided the S class into being aware and unaware of a disease, based on the SIS model, and concluded that media campaigns are helpful in decreasing the spread of infectious diseases by ensuring that individuals are more cautious of making contact with those that are infected.

Another way of representing the media effect in a mathematical model is reducing the transmission rate. This is based on the assumption that individuals who are aware of the disease will change their behavior, causing the transmission rate to decrease. There have been a few ways of reducing the transmission rate. Cui et al. [5] and Sun et al. [10] reduced the transmission rate by lowering the contact rate term in the model. Other studies utilized a transmission rate that decays exponentially as a function of the number of infected people [4]. In other words, the media reports only the number of infected people and the transmission rate decays exponentially as the reported number of infected people increases. In the model of Liu et al. [8], in contrast, the exponential decay is a function of the number of those exposed and hospitalized as well as the number of infected people; the transmission rate decreases exponentially as the media-reported numbers of exposed, hospitalized, and infected people increase. Most models considered only the number of infected individuals in the media effects, however, both work [11, 12] made a more realistic assumption that people tend to be sensitive not only to the number of infected people but also to whether diseases spread is getting better or worse. Thus, they built a model where the transmission rate is a function of both the number of infected

people and its rate of change. This media term will be employed as our theory-based model in the next section.

Also, a non-linear incidence rate was utilized to represent the media effect in a mathematical model. Tchuenche et al. [12, 13] incorporated the contact rate reduction parameters, which represent the change in human behavior when the numbers of infectious and vaccinated individuals are reported in the media. A similar approach was used by Li and Cui [14], with the only difference being that they only took into account the number of infectious individuals. In a similar vein, Xiao and Ruan [15] adopted a nonmonotonic incidence rate that rapidly decreases as the number of infected individuals increases. They considered this as a psychological effect based on the assumption that people tend to reduce the contact rate when there are many infected people in the population. This psychological effect can be regarded as identical to the media effect because the only practical way that an individual can receive information about the number of infected people is through the media. The model in the work of Mummert and Weiss [16] was based on the assumption that the media would report the number of new infections and deaths.

In addition, some studies explored the influence of the time gap between awareness and vaccination behavior. d'Onofrio et al. [17, 18] incorporated the notion of decaying memory based on the assumption that individuals do not necessarily get vaccinated immediately after they learn about the infectious disease. Buonomo et al. [19] paid attention to the fact that individuals sometimes give more weight to past information and incorporated this into their mathematical model.

As seen from the above, there has been much literature about how the media affects the spread of infectious disease. Yet almost no research has been done trying to fit this model to real-world media coverage data. Yoo et al. [20], Ma et al. [21] and Reintjes et al. [22] statistically analyzed the relationship between media coverage and vaccination rate, but they did not explore how those two dynamically affect each other or the resulting dynamics of infectious disease transmission.

This study explores how media coverage influences the infection dynamics. Therefore, we focus on the case of H1N1 in 2009, South Korea. The media coverage data were gathered from the news article database, KINDS (www.kinds.or.kr), maintained by the Korea Press Foundation using the query term 'H1N1' or 'novel flu' (English translation of informal Korean words for H1N1) and counting the number of news articles that matched the query term [23]. The 2009 H1N1 confirmed cases were taken from the Korea Centers for Disease Control and Prevention [24]. Both of the data sets span from April 2009 to August 2010 and are transformed into a week-based format. A mathematical model in [25] is modified by incorporating the media effect. The present study utilizes two kinds of models: one incorporates a theory-based media effect term from previous research [11, 12] and the other incorporates a data-based media effect term from media coverage data during the 2009 H1N1 season in South Korea. Both of them are fitted to the incidence data in the same period of epidemic duration. Using these two models, the relationship between media coverage and influenza dynamics is explored.

## 2 An influenza transmission model with the media effect

### 2.1 Modeling the media effect

We propose a mathematical model to investigate the media effects on the influenza transmission dynamics of H1N1 in South Korea, 2009. Our mathematical model is based on the model in the previous study [25]. A standard compartment model is employed to divide the population into the following compartments with different epidemiological status; susceptible ($S$),

vaccinated ($V$), exposed ($E$), clinically ill and infectious ($I$), asymptomatic but still infectious ($A$), hospitalized ($H$), recovered ($R$) and cumulative clinically ill and infectious cases ($C$).

Since the disease-induced death is negligible for a single outbreak duration (less then a year), it is assumed that the total population size, $N(t)$, remains constant, where $N(t) = S(t) + V(t) + E(t) + I(t) + A(t) + H(t) + R(t) \equiv N$. In addition, the population is assumed to be completely susceptible at the beginning of the epidemic. $\beta$ is the baseline transmission rate and $\eta$ stands for the relative infectiousness of asymptomatic cases compared with symptomatic cases ($0 \leq \eta \leq 1$), $u$ is the vaccination rate with which the vaccination is administered to susceptible individuals, and $\sigma$ is the vaccine efficacy based on the assumption that the vaccine provides only partial immunity so that vaccinated individuals are less susceptible than unvaccinated individuals. Exposed individuals proceed to the infectious status after some latent period, which is represented as $k$ in the model, and a proportion $p$ of infected individuals become symptomatic. Infectious individuals are assumed to be hospitalized at the rate of $\alpha$, and are treated with an antiviral drug at the rate $f$. Both symptomatic and asymptomatic individuals recover at the rate $\gamma$. Hospitalized individuals recover at the rate $\theta$. Recovered individuals are assumed to gain immunity for the duration of the epidemic. The system of differential equations that describes our influenza transmission model is given as follows:

$$
\begin{aligned}
\frac{dS}{dt} &= -e^{-cM(t)}\beta\{\tfrac{\eta A + I}{N}\}S - uS, \\
\frac{dV}{dt} &= uS - (1-\sigma)e^{-cM(t)}\beta\{\tfrac{\eta A + I}{N}\}V, \\
\frac{dE}{dt} &= (S + (1-\sigma)V)e^{-cM(t)}\beta\{\tfrac{\eta A + I}{N}\} - kE, \\
\frac{dI}{dt} &= kpE - (\alpha + \gamma + f)I, \\
\frac{dA}{dt} &= k(1-p)E - \gamma A, \\
\frac{dH}{dt} &= \alpha I - \theta H, \\
\frac{dR}{dt} &= \gamma(A + I) + fI + \theta H, \\
\frac{dC}{dt} &= kpE.
\end{aligned}
\tag{1}
$$

As shown in the model, it is assumed that susceptible individuals become infected at the following rate:

$$
e^{-cM(t)}\beta\{\frac{\eta A + I}{N}\},
$$

which suggests that individuals adopt behaviors that may reduce the probability of being infected ($e^{-cM(t)}$) in $\beta$. The model in [25] is modified by incorporating the media effect in $\beta$ is the baseline transmission rate and $c$ is the nonnegative weight constant (a level of media coverage). The media effect is incorporated as the term $M(t)$, which is described below.

In the present study, we consider two scenarios to incorporate the media effect into our influenza transmission model.

(M1). Model 1

$M(t) = max\{0, aI(t) + b\frac{dI}{dt}(t)\}$, where the media effect is the term $M(t)$, which is the sum of the number of infected individuals and its rate of change with scale constants $a$

and $b$, respectively. It is based on the assumption that people will pay attention both to the current situation of influenza and whether it is getting better or worse, as in [11, 12].

(M2). Model 2

$M(t) = d \times m(t)$, where $m(t)$ is the amount of media coverage at time $t$ and $d$ is a scale constant ($m(t)$ and $d$ are nonnegative).

In Model 2, the media coverage data $m(t)$ were gathered from the news article database, KINDS (www.kinds.or.kr), maintained by the Korea Press Foundation using the query term 'H1N1' or 'novel flu' (English translation of informal Korean words for H1N1) and counting the number of news articles that matched the query term [23]. The 2009 H1N1 confirmed cases were taken from the Korea Centers for Disease Control & Prevention [24]. Both of the data span from April 2009 to August 2010 and are transformed into a week-based format.

In Fig 1, time series of H1N1 cases is displayed in the top panel. Media coverage data during the same period is displayed in the bottom panel. Note that the index case of H1N1 was traced back during April 20-26, 2009 and confirmed in May 1, 2009 [26]. Due to this index case, the number of articles on H1N1 increased dramatically from May, 2009 (see the number of new articles in the bottom panel: 1 in the first week, 2 in the second week, 0 in the third week, and 43 in the fourth week). It is clear that Korean press began to pay attention to H1N1 only after confirmation of the first case in Korea. This leads that the first peak of the media data in May, 2009 before the actual H1N1 peak (in October, 2009).

## 2.2 The basic reproduction number

The basic reproduction number is defined as the average number of secondary cases generated by a primary infectious case in a completely susceptible population [27]. The basic reproduction number for our influenza model (1) is computed by the next-generation method [28]. The model (1) can be rearranged, and its solutions can be expressed by $\mathbf{x} = (x_1, \cdots, x_7) = (E, I, A, S, V, H, R)$. Let the right side of the system (1) be zero and one can verify that $\mathbf{x}_0 = (0, 0, 0, N, 0, 0, 0)$ is the equilibrium. We refer to this equilibrium as the disease-free equilibrium (DFE). Then, the disease-free equilibrium (DFE) is used to evaluate the basic reproduction number. The basic reproduction number, denoted $\mathcal{R}_0$, can be evaluated using the two vectors $\mathcal{F}$ and $\mathcal{V}$ defined as:

$$
\mathcal{F} = \begin{bmatrix} S\beta\{\frac{\eta A + I}{N}\}e^{-cM(t)} \\ 0 \\ 0 \\ 0 \\ 0 \\ 0 \\ 0 \end{bmatrix}, \quad
\mathcal{V} = \begin{bmatrix} -(1-\sigma)\beta V e^{-cM(t)}\{\frac{\eta A + I}{N}\} + kE \\ -kpE + (\alpha + \gamma + f)I \\ -k(1-p)E + \gamma A \\ \beta S e^{-cM(t)}\{\frac{\eta A + I}{N}\} + uS \\ (1-\sigma)e^{-cM(t)}\beta V\{\frac{\eta A + I}{N}\} - uS \\ -\alpha I + \theta H \\ -\gamma(A + I) - fI - \theta H \end{bmatrix}.
$$

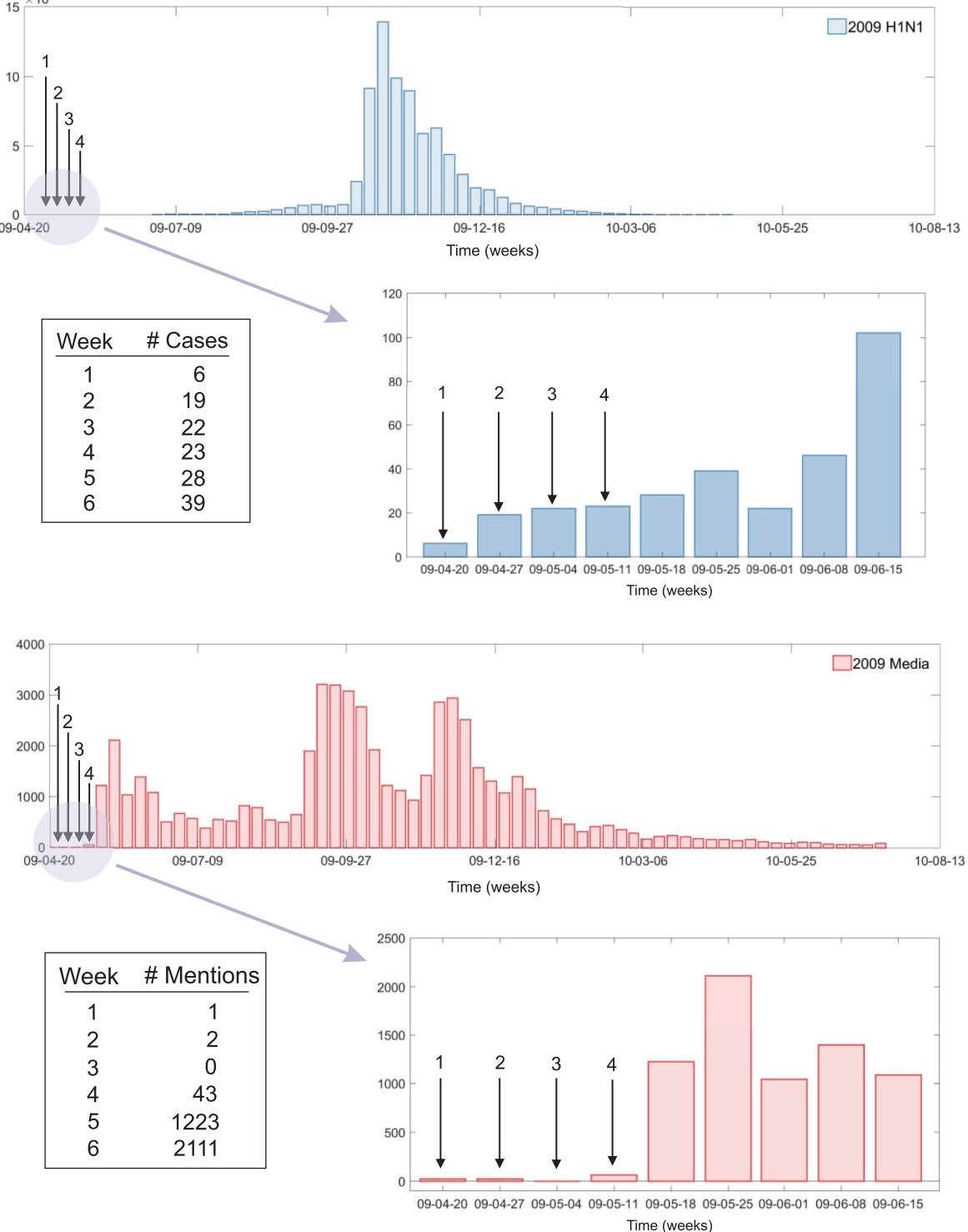

**Fig 1. Time series of H1N1 cases is displayed from April 20, 2009, to July 18, 2010, in the top panel.** Media coverage data during the same period are displayed in the bottom panel. The amount of media coverage dramatically increased in May, 2009, shortly after the first case of H1N1 in South Korea was appeared on April 20-26, 2009.

We take the Fréchet derivatives of $\mathcal{F}$ and $\mathcal{V}$ and evaluate them at the DFE ($F = \left(\frac{\partial \mathcal{F}}{\partial x_i}\right)$ and $V = \left(\frac{\partial \mathcal{V}}{\partial x_i}\right)$). Then, we obtain

$$
F = \begin{bmatrix} 0 & \beta & \beta\eta \\ 0 & 0 & 0 \\ 0 & 0 & 0 \end{bmatrix}, V = \begin{bmatrix} k & 0 & 0 \\ -kp & (\alpha+\gamma+f) & 0 \\ -k(1-p) & 0 & \gamma \end{bmatrix}.
$$

$F$ is non-negative and $V$ is a non-singular matrix with $V^{-1}$

$$
V^{-1} = \begin{bmatrix} \frac{1}{k} & 0 & 0 \\ \frac{p}{(\alpha+\gamma+f)} & \frac{1}{(\alpha+\gamma+f)} & 0 \\ \frac{1-p}{\gamma} & 0 & \frac{1}{\gamma} \end{bmatrix}.
$$

Thus, $FV^{-1}$ is non negative

$$
FV^{-1} = \begin{bmatrix} \frac{\beta p}{(\alpha+\gamma+f)} + \frac{\beta\eta(1-p)}{\gamma} & \frac{\beta}{(\alpha+\gamma+f)} & \frac{\beta\eta}{\gamma} \\ 0 & 0 & 0 \\ 0 & 0 & 0 \end{bmatrix}.
$$

Hence, the basic reproduction number is the spectral radius of $FV^{-1}$ given as

$$
\mathcal{R}_0 = \frac{\beta p}{(\alpha+\gamma+f)} + \frac{\beta\eta(1-p)}{\gamma}. \tag{2}
$$

It is assumed that the population is completely susceptible at the beginning of the epidemic and the $\mathcal{R}_0$ measures the average secondary cases at the beginning of the epidemic. This implies that media coverage is assumed to have no effect at the beginning of the epidemic. Therefore, media coverage does not play a role in the basic reproduction number $\mathcal{R}_0$ in Eq (2). Note that the basic reproduction number $\mathcal{R}_0$ is a function of six parameters. The basic reproduction number $\mathcal{R}_0$ includes some intervention parameters: a treatment rate ($f$) and a hospitalization rate ($\alpha$). Let us denote the basic reproduction number by $\mathcal{R}_0$ when there are no interventions, that is, $f = 0$ and $\alpha = 0$. We will distinguish $\mathcal{R}_0$ with the controlled reproduction number $\mathcal{R}_c$ when there are interventions ($f > 0$ and $\alpha > 0$).

## 3 Numerical results

### 3.1 The basic reproduction number

We present numerical simulations to explore the effects of various parameters on the basic reproduction number $\mathcal{R}_0$. Influenza parameters are taken from [25], which investigated the 2009 H1N1 influenza dynamics in Korea. These baseline parameter values are collected in Table 1. Let us recall that a proportion $0 < p < 1$ of exposed individuals progress to the infectious class I at the rate k while the rest $(1 - p)$ progress to the asymptomatic partially infectious class A at the same rate $k$. Here, $\eta$ is the relative constant ($0 < \eta < 1$) that accounts for the reduction in transmissibility for asymptomatic individuals (partially infectious). Note that $p$ and $\eta$ are associated with asymptomatic individuals and they have higher uncertainty levels (larger variances) than other parameter values as investigated in [29].

**Table 1. Baseline parameter values taken from the previous work [25].**

| Parameter | Description | Value |
|---|---|---|
| $\sigma$ | Vaccine efficacy | 0.8 |
| $\eta$ | Relative infectiousness of asymptomatic cases | 0.142 |
| $k$ | Rate of progression from the latent to infected (days$^{-1}$) | 0.833 |
| $p$ | Proportion of infected individuals who become symptomatic | 0.33 |
| $\gamma$ | Recovery rate for infected individuals (days$^{-1}$) | 0.22 |
| $\theta$ | Recovery rate for hospitalized individuals (days$^{-1}$) | 0.34 |
| $u(t)$ | Vaccination rate (days$^{-1}$) | 0-0.006 |
| $f(t)$ | Antiviral rate (days$^{-1}$) | 0-0.6 |
| $\alpha(t)$ | Diagnostic rate (days$^{-1}$) | 0-0.08 |
| $c$ | Level of media coverage | 1 |
| $S(0)$ | Initial number of susceptible individuals | $4.8746693 \times 10^7$ |
| $I(0)$ | Initial number of infected individuals | 1 |

First, we carry out sensitivity of $\mathcal{R}_0$ with varying each of six individual parameters. Fig 2 illustrates a sensitivity analysis of $\mathcal{R}_0$ to the variation of six parameters; parameters are made to vary in the following ranges; $\beta = [0.3, 1.5]$, $p = [0.1, 0.9]$, $\gamma = [0.1, 0.5]$, $\eta = [0.1, 0.5]$, $f = [0.1, 0.5]$, $\alpha = [0.1, 0.5]$. The sensitivity is carried out as we vary each parameter while other parameters are kept as baseline values in Table 1. Recall that the basic reproduction number $\mathcal{R}_0$ is a function of six parameters as shown in (2); $\mathcal{R}_0$ has a linear relationship with $\beta$, $p$, and $\eta$. Therefore, the basic reproduction number increases for larger values of $\beta$, $p$ or $\eta$ in a linear fashion. Similarly, the basic reproduction number has a reciprocal relationship with $f$, $\alpha$, and $\gamma$; it

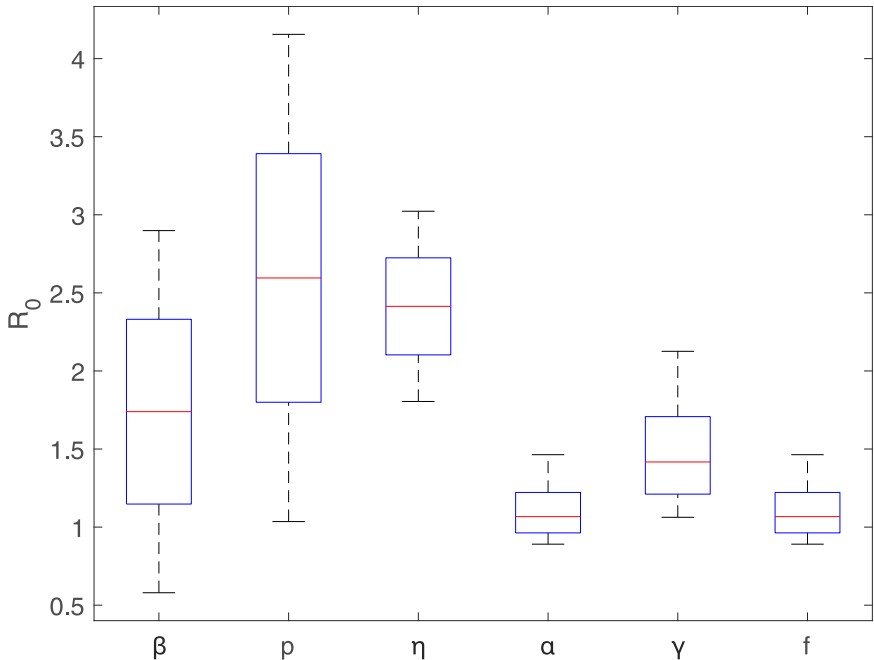

**Fig 2. Sensitivity of $\mathcal{R}_0$ with varying each individual parameter; $\beta = [0.3, 1.5]$, $p = [0.1, 0.9]$, $\eta = [0.1, 0.5]$, $\gamma = [0.1, 0.5]$, $f = [0.1, 0.5]$, $\alpha = [0.1, 0.5]$ (the baseline parameter values are given in Table 1).**

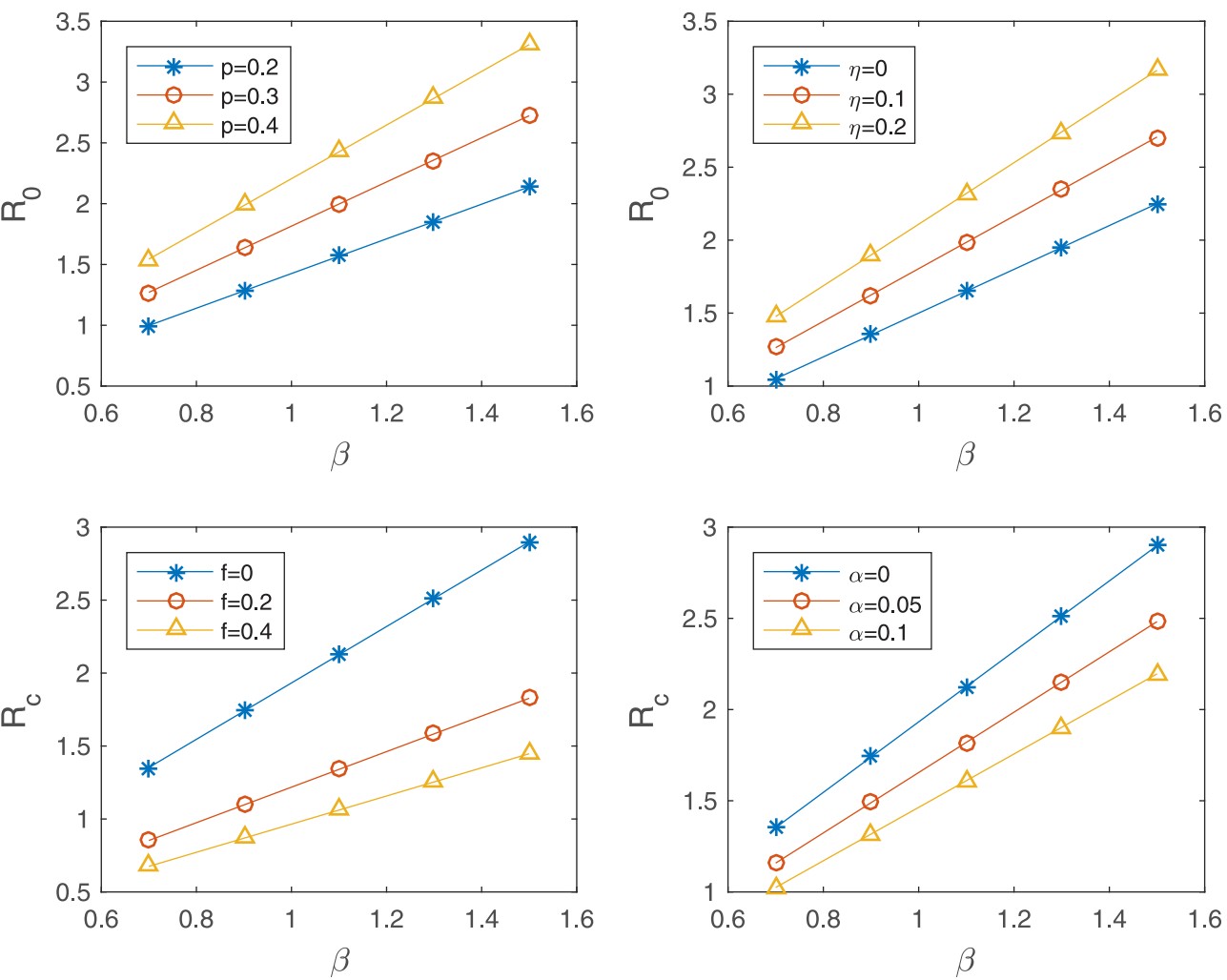

**Fig 3.** Sensitivity analyses for $\mathcal{R}_0$: (a) The impact of $p$ when $\gamma = 0.22$, $\eta = 0.142$, $f = 0$, $\alpha = 0$, (b) the impact of $\eta$ when $\gamma = 0.22$, $p = 0.33$, $f = 0$, $\alpha = 0$; sensitivity analyses for $\mathcal{R}_c$: (c) the impact of $f$ when $\gamma = 0.22$, $\eta = 0.142$, $p = 0.33$, $\alpha = 0$, (d) the impact of $\alpha$ when $\gamma = 0.22$, $\eta = 0.142$, $f = 0$, $p = 0.33$.

decreases as $\gamma$, $f$ and $\alpha$ increases. Also, the $\mathcal{R}_0$ shows the larger ranges for $\beta$, $p$ or $\eta$ due to their higher uncertainty levels.

Next, the effects of the inclusion of the asymptomatic class and the reduction constant of the A-class are further explored, as well as the effects of interventions of treatment ($f$) and hospitalization ($\alpha$). The impacts of the three parameter values $\beta$, $p$, $\eta$ are $f$ and $\alpha$ on the basic reproduction number $\mathcal{R}_0$ are displayed in Fig 3. The results are shown as a function of the baseline transmission rate ($\beta$) for each parameter. For larger values of $p$ and $\eta$, the basic reproduction number increases as the baseline transmission rate ($\beta$) increases (top panels). In the presence of interventions, the controlled reproduction number $\mathcal{R}_c$ becomes smaller under the implementation of intensive (higher) treatment and hospitalization (bottom panels). Overall, the basic (or controlled) reproduction number gets larger (smaller) as $p$ or $\eta$ (or $f$ and $\alpha$) get larger in a straightforward manner.

Furthermore, Fig 4 shows three-dimensional results: the impact of the two parameter values $p$ and $\eta$ ($f$ and $\alpha$) on the basic (controlled) reproduction number. Similarly, the overall basic reproduction number $\mathcal{R}_0$ increases as $p$ and $\eta$ increase (see Fig 4(a)). Again, the controlled

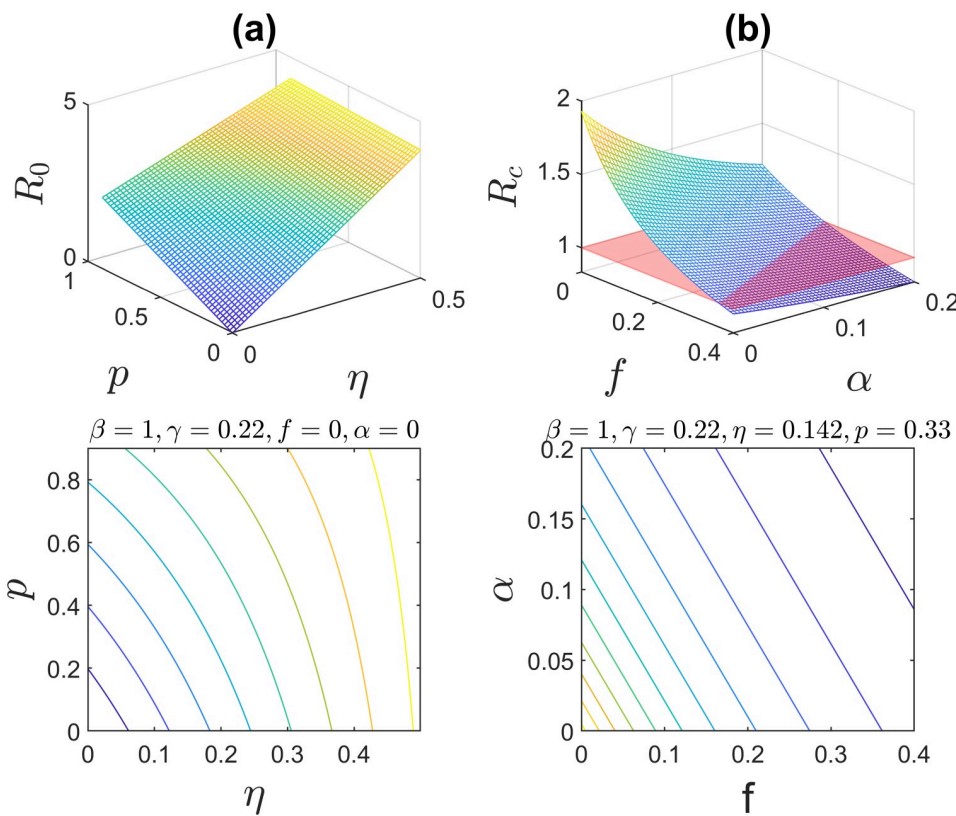

**Fig 4.** (a) Sensitivity analyses of the basic reproduction number for $p$ and $\eta$ ($\beta = 1$, $\gamma = 0.22$, $\alpha = f = 0$). (b) Sensitivity analyses of the controlled reproduction number for $\alpha$ and $f$ ($\beta = 1$, $\gamma = 0.22$, $p = 0.33$, $\eta = 1.42$). The bottom panels are the contour plots of (a) and (b), respectively.

reproduction $\mathcal{R}_c$ becomes smaller under more intensive treatment and hospitalization, as shown in Fig 4(b).

## 3.2 Parameter estimation

In this section, we estimate parameters for the media effect term under two models, whose only difference lies in the media effect term, $M(t)$. This media term is incorporated into the incidence rate, $e^{-cM(t)} \beta \{\frac{\eta A + I}{N}\}$. Model 1 incorporated the media effect term as $M(t) = max\{0, aI(t) + b\frac{dI}{dt}(t)\}$. Model 2 utilized the real-world media coverage data of H1N1 in South Korea 2009 as $M(t) = d \times m(t)$. Here, $m(t)$ is the amount of media coverage of H1N1 at time $t$ and $d$ is the scale constant ($m(t)$ is obtained from [23]). Both models were fitted to the actual 2009 H1N1 data provided by KCDC (the cumulative number of confirmed cases during April 2009 to August 2010) [24].

For Model 1, the transmission rate $\beta$ and the scale constants $a$, $b$ in the media term were estimated. For Model 2, the transmission rate $\beta$ and the scale constant $d$ in the media term were estimated. The rest of the parameter values were fixed, as given in Table 1. Initial conditions for other variables are set to be zero ($V(0) = E(0) = A(0) = H(0) = R(0) = C(0) = 0$). The three parameter values $\hat{\Theta}_1 = (\beta, a, b)$ in Model 1 and the two parameter values $\hat{\Theta}_2 = (\beta, d)$ in Model 2 were estimated through least-squares fitting of $C(t)$ (the last differential equation for

the cumulative number of confirmed cases) in system (1). Denote the observed cumulative cases data in day as $Y = (y_1, \cdots, y_M)$ and the influenza model output for $C(t)$ as $C(m, \Theta)$ with $(m = 1, \cdots, M)$. The parameter vector $\hat{\Theta}$ for both models was obtained by minimizing $argmin_{\Theta \in W} J(\Theta)$ for a cost functional $J(\Theta) = ||Y - C(\cdot, \Theta)||^2 = \sum_{m=1}^{M} |y_m - C(m, \Theta)|^2$, where $W = \{\Theta \in D | lb \leq \Theta \leq ub\}$ was an admissible set for the parameter vector $\Theta$. Note that $D$ is a $n$-dimensional real vector space; $D = R^3$ for Model 1 and $D = R^2$ for Model 2.

As shown in Fig 1, the intrinsic nature of 2009 H1N1 epidemic curves included the following four stages: very small cases, a tiny first peak, a second huge peak and very small cases again. Using only one parameter set in the entire time window, we had a very crude approximation of the model fit. Therefore, for the better fit to the data, the entire time interval ([1, 455] days) was divided into the following four subintervals: $Period_1 = [1, 65]$, $Period_2 = [65, 170]$, $Period_3 = [170, 275]$, $Period_4 = [275, 455]$ (in days).

Both $\hat{\Theta}_1$ and $\hat{\Theta}_2$ were estimated for Model 1 and Model 2 in a sequential manner. First, $\hat{\Theta}_1$ ($\hat{\Theta}_2$) was estimated in $Period_1$, then $\hat{\Theta}_1$ ($\hat{\Theta}_2$) was estimated in $Period_2$ using the initial conditions (which were obtained from the model output values at $65day$). We completed this sequential process for $Period_3$ and $Period_4$, respectively. For the least-squares fitting (numerical) procedure on each subinterval, we employed the Trust–Region-Reflective method implemented in MATLAB (The Mathworks, Inc.) in the built-in routine *fimincon*, which was part of the optimization toolbox (with constraints on positive parameter values).

This is a typical inverse problem associated to many epidemic models. Generally, it is highly nonlinear, with none or low regularity, multiple solutions (or even none), being very complicated to solve [30, 31]. The inverse problem seeks to estimate a finite number of parameters on a finite dimensional space with a nonlinear cost function. Furthermore, Theorem 4.5.1 in [32] showed a proper sense of well-posedness of the inverse problem (existence, uniqueness and local stability). Also, the Trust-Region-Reflective method (TRR) is employed here to numerically approximate a solution for the inverse problem [33]. Hence, this ensures the global minimum of the parameter estimation under the above assumptions. More details of theses inverse problems are given in [34].

As shown in Fig 5, our fitted model for Model 1 and Model 2 during the 2009 influenza epidemic duration in Korea agreed well with the observed epidemic data. The resulting parameter estimates of the four-time windows for both models were collected in Table 2. The numbers of cumulative cases under Model 1 and Model 2 (solid curves) were compared with the actual cumulative 2009 H1N1 data (circle): the top panel for Model 1 and the bottom panel for Model 2. Dashed vertical lines divided the entire time window into the four subintervals.

### 3.3 The impact of media on influenza dynamics

In this section, we investigate the impacts of the media effect term on influenza transmission dynamics under Model 1 and Model 2. We carry out numerical simulations using the estimated parameters above and the rest of the baseline parameter values in Tables 1 and 2. First, Fig 6(a) illustrates the resulting dynamics of Model 1 whose media effect term is based on theory. Fig 6(b) illustrates the dynamics of the number of infected individuals and its rate of change, both of which are elements of $M(t)$ in Model 1. Next, Fig 7(a) illustrates the resulting dynamics of Model 2, whose media effect term is based on the real-world media coverage data during the 2009 H1N1 epidemic duration. The media term $M(t) = d \times m(t)$ is displayed in Fig 7(b). The overall results are almost identical to the results in Fig 6, which means that the media effect term in Model 1 effectively captures the media effect that manifests in real-world data.

More detailed dynamics of the two media effect terms are compared in Fig 8: Model 1 (solid) and Model 2 (dashed). Under the parameters we obtained in this study, the two

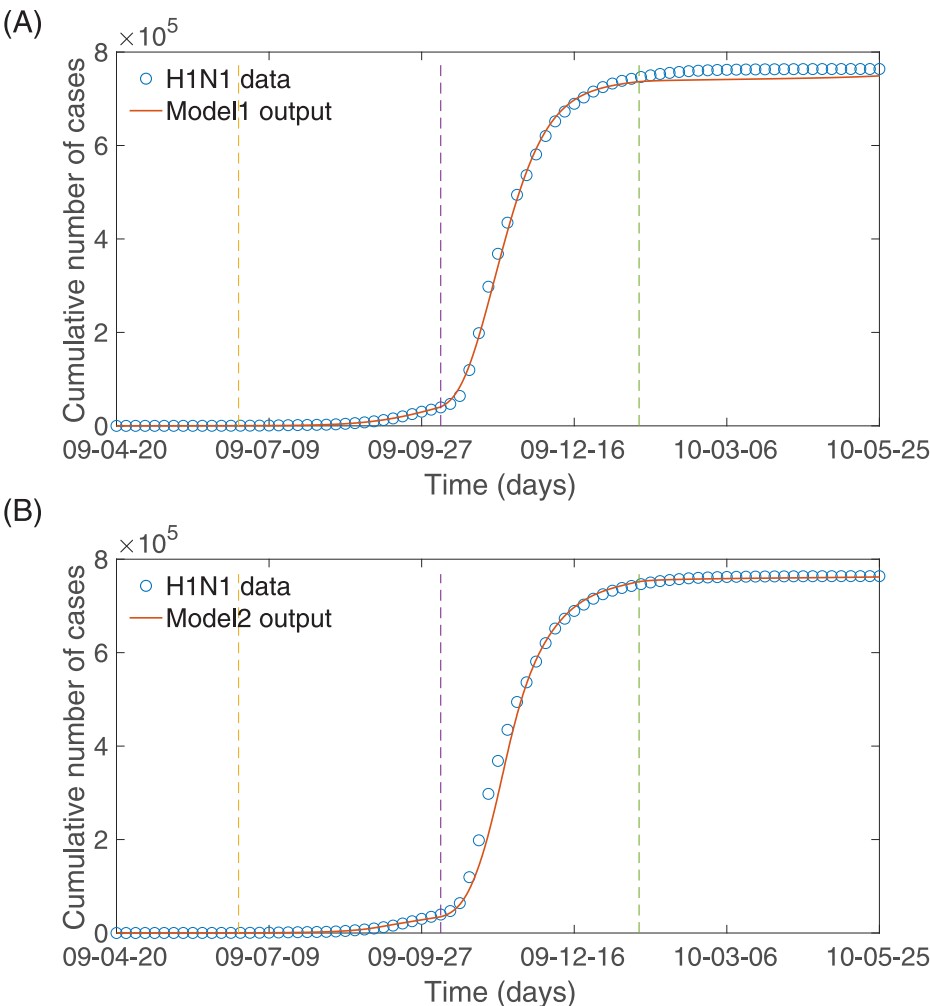

**Fig 5. The best-fit cumulative number of cases obtained by fitting $C(t, \hat{\Theta})$ (solid curve) in system (1) to the cumulative number of H1N1 cases (circle): The top panel for Model 1 and the bottom panel for Model 2.**

resulting dynamics are very similar with double peaks as observed in the media coverage data (the bottom panel of Fig 1). Note that the first peak is larger than the second peak in both media terms. The first peak appeared right before the actual epidemic peak. Interestingly, the actual H1N1 peak appeared in the middle of double media peaks. This has been observed in other research [21, 22]: the actual media data showed the dip during the time where the actual epidemic peak occurred. Even though their data showed a similar dip of media coverage

**Table 2. Estimated parameter values.**

|  | Model 1 | | | Model 2 | |
|---|---|---|---|---|---|
|  | $\beta$ | a | b | $\beta$ | d |
| Period 1 | 0.5797 | 7.5661e-13 | 2.3335e-14 | 0.5701 | 9.7969e-13 |
| Period 2 | 0.7201 | 1.4453e-04 | 1.1071e-8 | 0.7566 | 1.2172e-04 |
| Period 3 | 1.1934 | 4.3325e-06 | 4.8625e-10 | 1.1704 | 6.6231e-05 |
| Period 4 | 0.7027 | 1.7356e-06 | 1.062e-04 | 0.7479 | 6.3447e-18 |

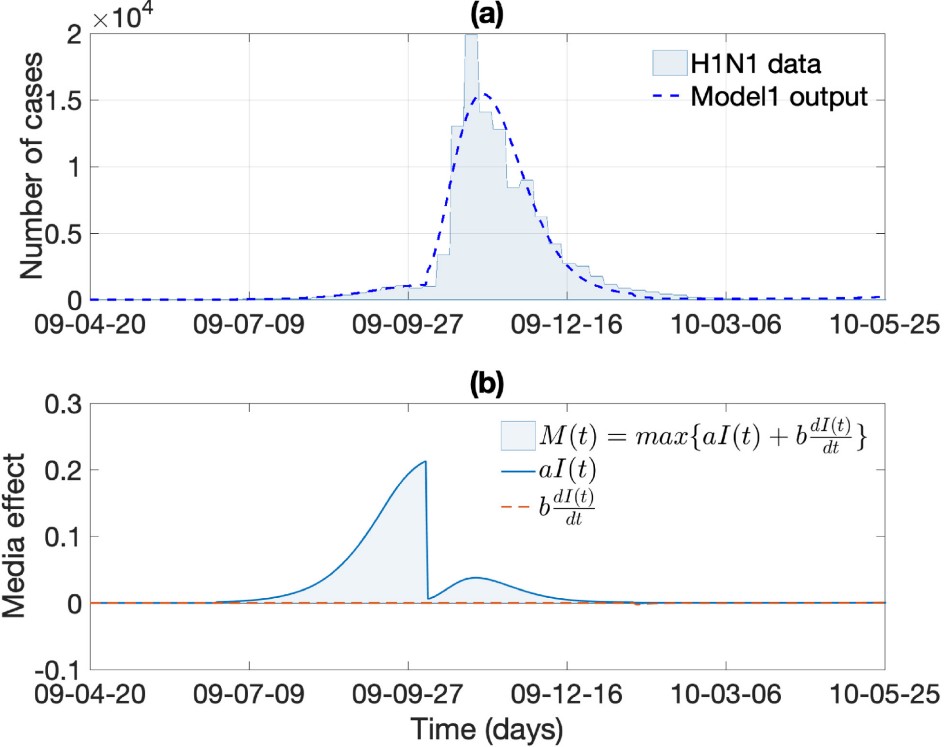

**Fig 6.** Model 1 output: (a) the best-fit solution obtained by fitting $C(t, \hat{\Theta})$ (dashed curve) in system (1) to the cumulative number of H1N1 cases is displayed. (b) Dynamics of the media effect term $M(t) = max\{0, aI(t) + b\frac{dI}{dt}(t)\}$.

during the influenza season, no clear explanations were addressed. As seen in Fig 1, the actual epidemic peak happened about five months after the first and second media peaks. This suggests that the media became loose the alert and thus reduced the amount of coverage. Soon, the epidemic peak occurred, and media coverage increased again. This is only a conjecture, and these issues should be addressed in future research.

Finally, the impacts of media coverage $c$ on influenza dynamics is investigated using three values of $c = 0.5, 1, 1.5$. Figs 9 and 10 illustrate incidence and cumulative incidence for Model 1 and Model 2, respectively. The higher the media coverage, the smaller the peak of epidemic curves in the results of both models. Specifically, the peak of epidemic curves would be significantly higher if there were less media coverage ($c = 0.5$) in comparison to the default ($c = 1$) amount of media coverage. As the value of $c$ increases ($c = 1.5$), which is equivalent to increasing the amount of media coverage, both the peak size and the final epidemic size decrease in a straightforward manner. The results of a more detailed analysis are presented in Table 3. In addition, the impact of media coverage on other classes is shown in Figs 11 and 12 for Model 1 and Model 2, respectively. Time series of S (susceptible), V (vaccinated), E (exposed individuals), A (asymptomatic individuals), H (hospitalized individuals), and R (recovered individuals) are displayed under three values of mass media coverage, $c$. Similarly, as the value of $c$ increases ($c = 1.5$), which is equivalent to increasing the amount of media coverage, both the peak size and the total size decrease in each class (E, A, H). The comparison above suggests that the theory-based and data-based media effect terms have almost the same influence on the influenza

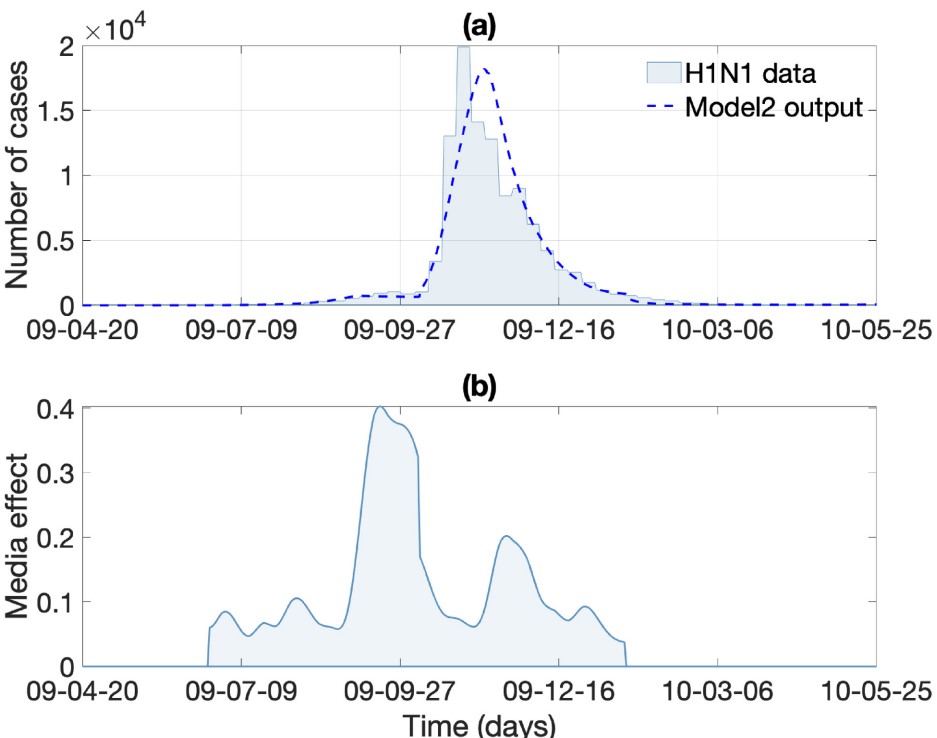

**Fig 7.** Model 2 output: (a) the best-fit solution obtained by fitting $C(t, \hat{\Theta})$ (dashed curve) in system (1) to the cumulative number of H1N1 cases is displayed. (b) Dynamics of the media effect term ($M(t) = d \times m(t)$) of Model 2 are displayed.

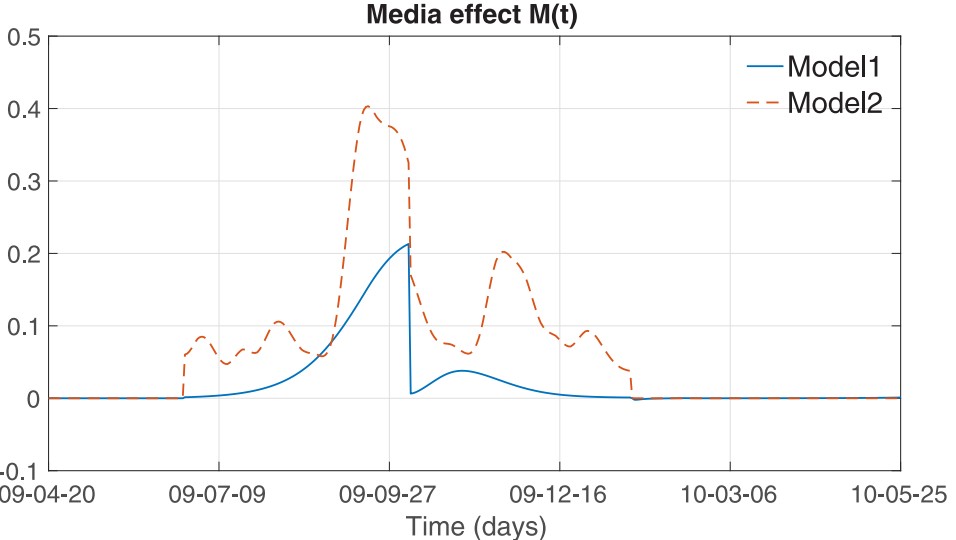

**Fig 8. Dynamics of the two media effect terms are compared: Model 1 (solid) and Model 2 (dashed).**

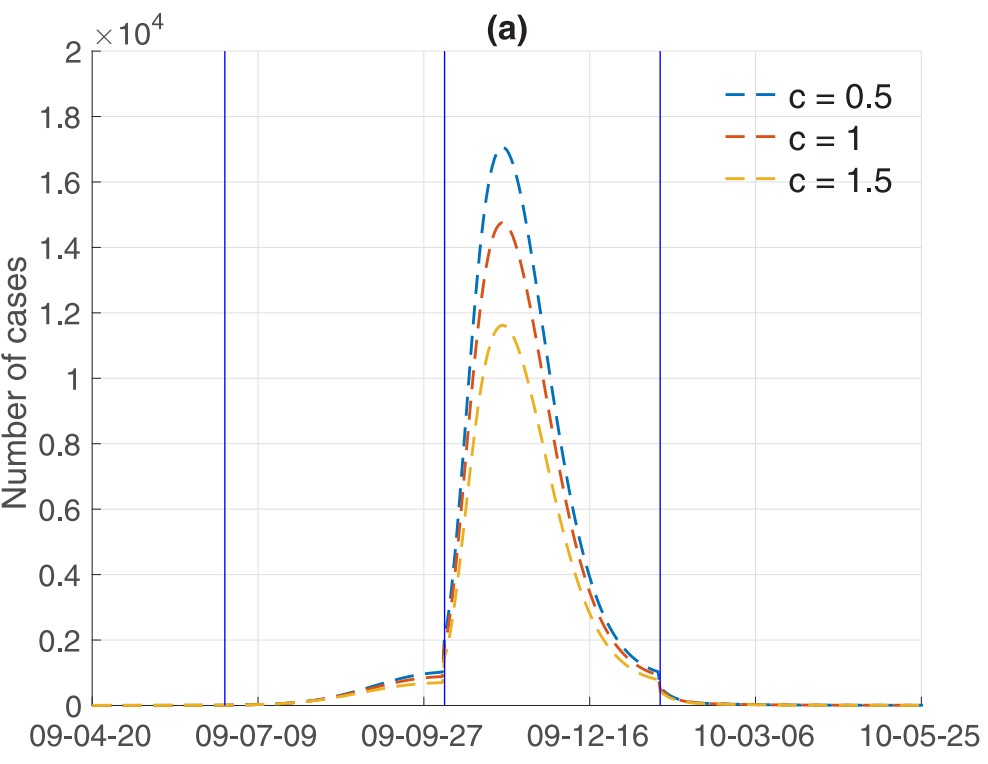

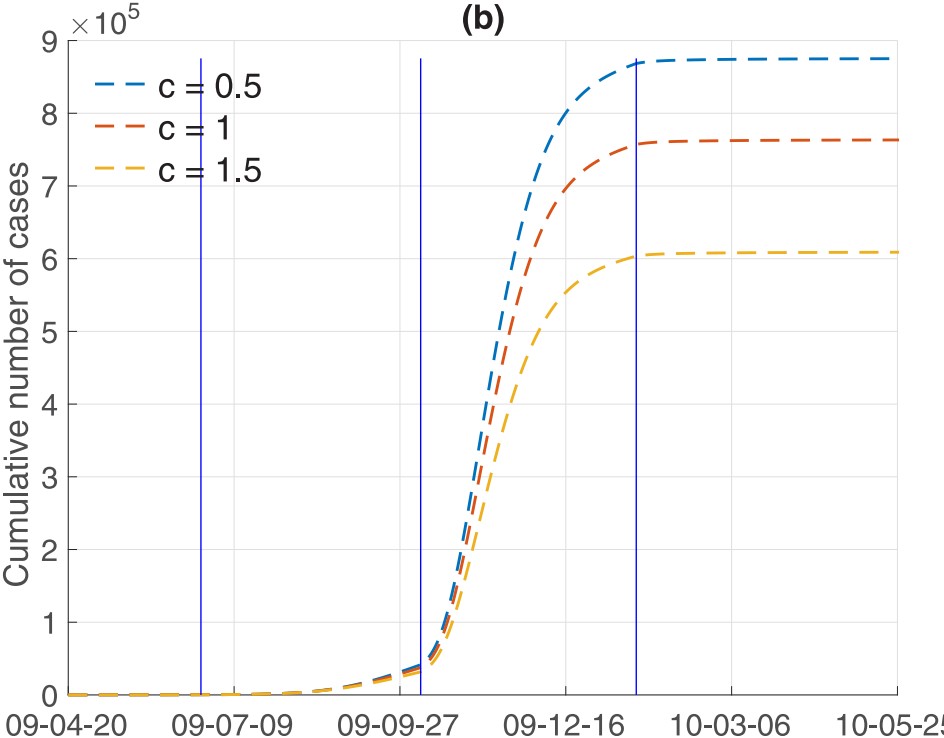

**Fig 9. Model 1 output: Incidence and cumulative incidence are displayed with varying degree of mass media coverage ($c$ = 0.5, 1, 1.5).**

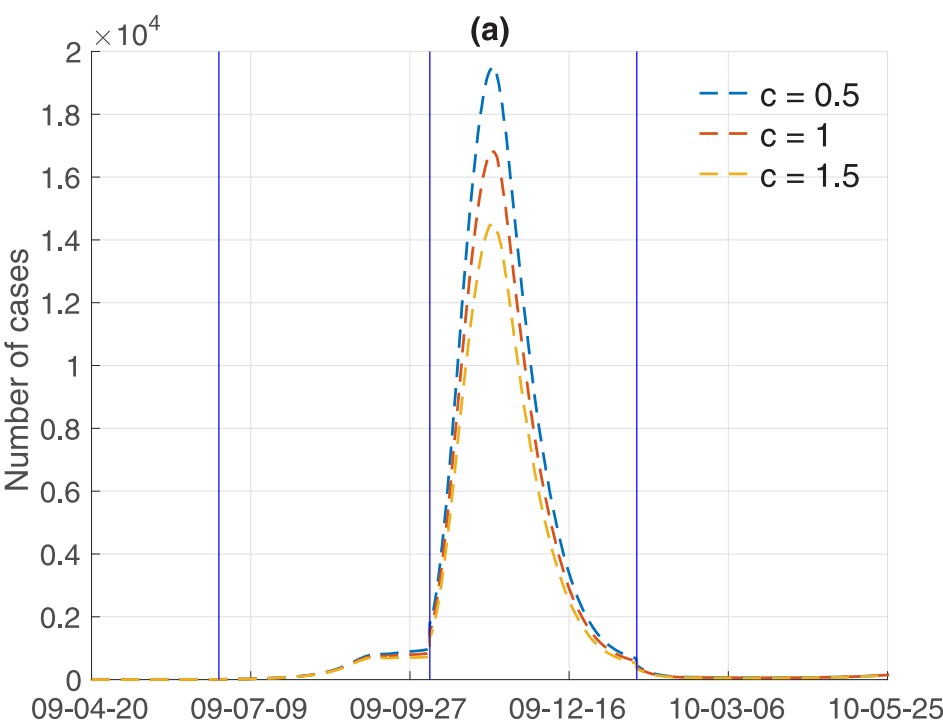

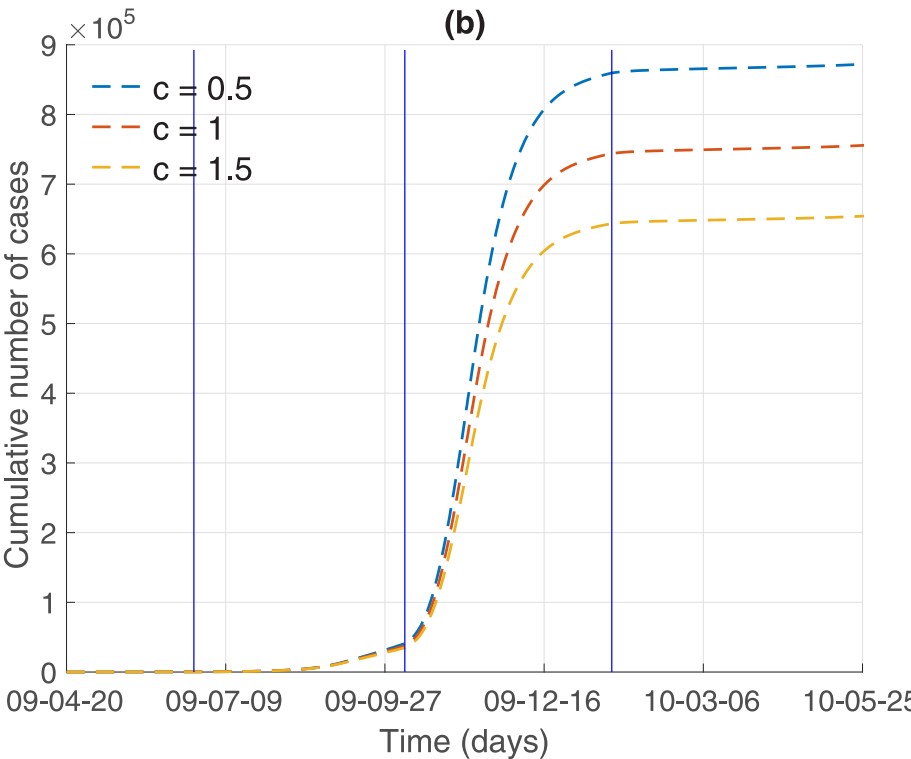

**Fig 10. Model 2 output: Incidence and cumulative incidence are displayed with varying degree of mass media coverage ($c$ = 0.5, 1, 1.5).**

**Table 3. Peak size and final epidemic size with varying degree of mass media coverage.**

| | | $c = 0.5$ | $c = 1$ | $c = 1.5$ |
|---|---|---|---|---|
| (a) Model 1 | Peak size | 17, 064 | 14,755 | 11,622 |
| | Proportion | 15.7% | - | -21.0% |
| | Cumulative | 892,637 | 763, 751 | 609,282 |
| | Proportion | 16.8% | - | -20.2% |
| (b) Model 2 | Peak size | 19, 491 | 16,837 | 14,504 |
| | Proportion | 15.5% | - | -13.8% |
| | Cumulative | 875,485 | 767, 561 | 675,010 |
| | Proportion | 14.0% | - | -12.0% |

dynamics under the parameters we obtained in this study. The final epidemic size shows that the results of Model 1 are slightly more sensitive to *c* (as shown in Table 3). This suggests that further modeling efforts need to be grounded in real-world knowledge.

## 4 Discussion

The media plays a crucial role in modern societies, which is also the case when it comes to infectious disease. As societies become larger and more diversified, an individual is hardly able to obtain enough correct information about the infectious disease in question for himself. Individuals inevitably depend on information that the media provide, and individuals alter their behavior based on that information. In this regard, it is proper to assume that media coverage might have an influence on infectious disease dynamics.

Concerning the media effect, previous studies have suggested a variety of results [4–7, 10, 17]. One of the key results is that the media effect by itself does not have an explicit effect on

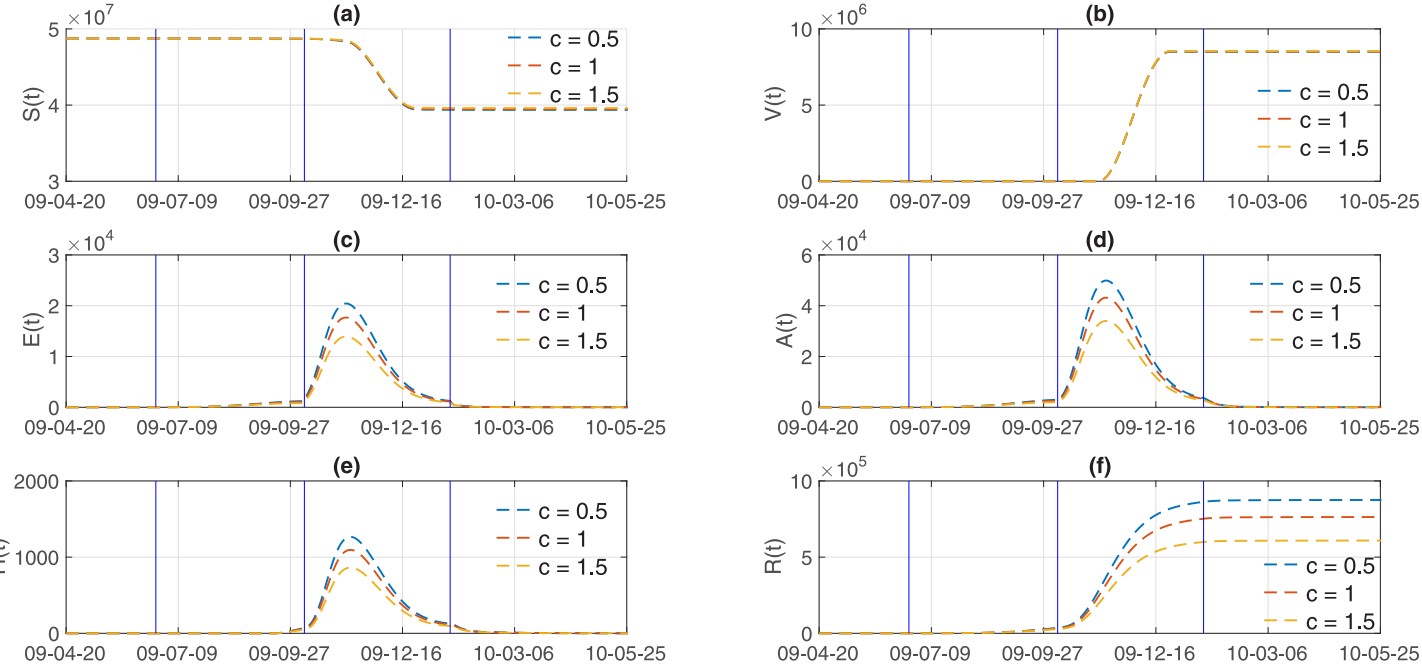

**Fig 11. Model 1 output: S (susceptible), V (vaccinated), E (exposed individuals), A (asymptomatic individuals), H (hospitalized individuals), and R (recovered individuals) are displayed with varying degree of mass media coverage (*c* = 0.5, 1, 1.5).**

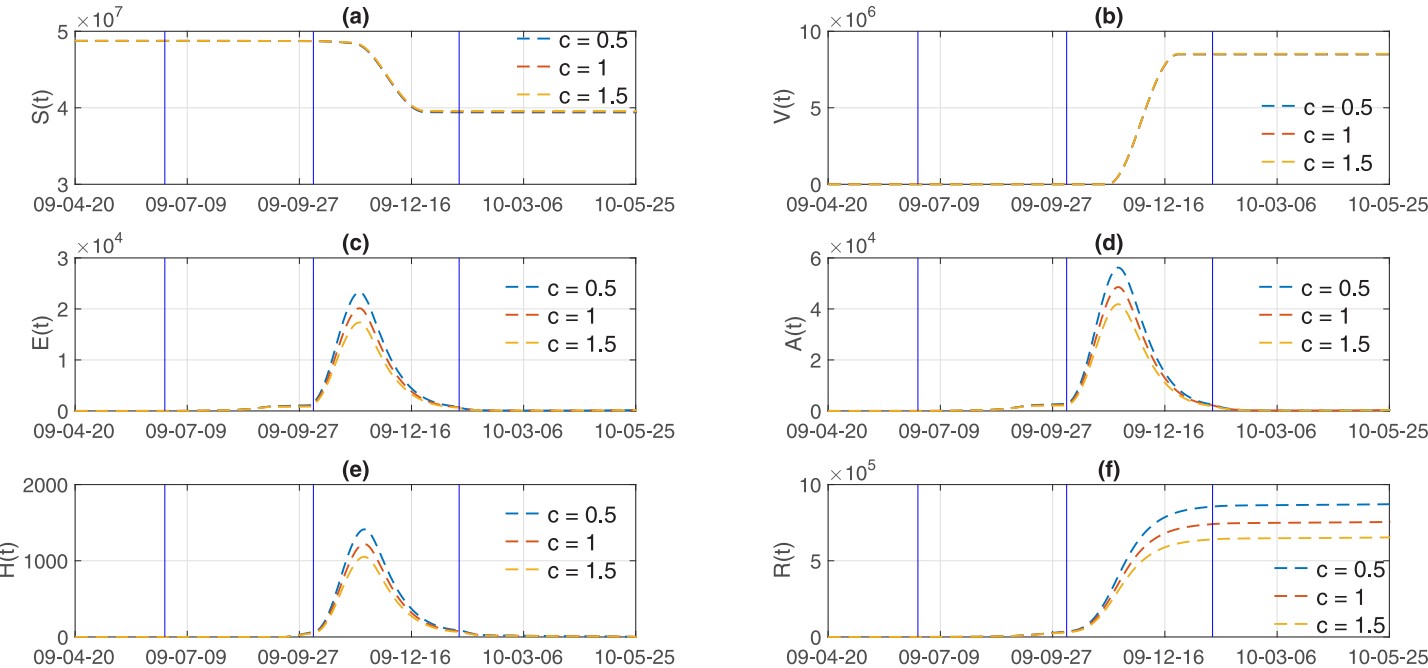

**Fig 12. Model 2 output: S (susceptible), V (vaccinated), E (exposed individuals), A (asymptomatic individuals), H (hospitalized individuals), and R (recovered individuals) are displayed with varying degree of mass media coverage ($c$ = 0.5, 1, 1.5).**

$\mathcal{R}_0$, thus mass media coverage alone cannot eradicate infectious disease [5, 9]. Instead, it can have different influences according to $\mathcal{R}_0$. When $\mathcal{R}_0$ is less than 1, the disease-free equilibrium is stable and the media does not have an explicit effect. When $\mathcal{R}_0$ is greater than 1, in contrast, the endemic equilibrium is stable and a higher level of media coverage may lead to a smaller level of infection prevalence [10, 16] or decelerate the spread of a disease [6, 7]. In addition, previous studies suggest that when to stop media coverage may be an issue; if the media stops coverage before the disease is eradicated, this may lead to another peak [5, 8]. Further, what information is given to the media can be an issue. Caution in reporting the decreased number of hospitalized individuals is important because it can lead to the misunderstanding among the people that the situation is getting better [8]. Further, if the mass media coverage began late after the outbreak of the disease, reporting current data is more helpful than reporting historical data [16].

In line with these studies, the present study investigated how the media affects influenza dynamics by incorporating the media effect term into the mathematical model. We estimated epidemiological parameters via least-squares fitting of the model to the cumulative number of the 2009 H1N1 data. What makes our study distinct is that the incorporated media effect term comes from the media coverage data as well as from theory based on previous studies. Results of numerical simulation suggest that the media can have a positive influence on influenza dynamics; more media coverage may lead to a reduced peak size and final epidemic size of influenza. Our results highlight that the theory-based and data-based media effect terms have almost the same influence on the influenza dynamics under the parameters we obtained in this study. However, the results can be different for highly transmissible influenza. This suggests that further modeling efforts need to be grounded in real-world knowledge.

It becomes critical to understand the complex interplay between media attention, risk perception, behavior changes and the transmission dynamics of infectious diseases. More work can be done regarding the effect of the media on infectious diseases. First, the individual diversity of media credibility can be explored. Since it might be the case that individuals do not believe media coverage to the same degree, incorporating diverse media credibility of individuals might make the model more elaborate. Next, influence of the leader's opinion may be another topic worth studying. Mass media exerts its influence not only directly on individuals, but also through the opinion leaders who are more intelligent than the lay person and can deliver the content of the media to the population. Thus, it can be assumed that how cautious the leader is regarding their opinion on infectious disease may have an influence on the behavior of other individuals who are under the influence of the leader.

Furthermore, there are other important factors that can be considered and incorporated into a mathematical model such as various forms of mass/social media (audience rate in TV, newspaper, or web page) and the characteristics of infectious diseases (size or location). In addition, to investigate which media forms work best at reducing transmission dynamics, further research should be provided on sensitivity of media dependent risk perception and behavior changes. Finally, the behaviors of journalists can be considered. Journalists sometimes cover an issue as independent investigators, but it is much more common that they form groups and behave as a member of the group. Such group dynamics can be assumed to have an influence on the covering behavior of journalists, and as a result, on the behavior of individuals regarding infectious disease.

## Supporting information

**S1 Data.**
(XLSX)

## Author Contributions

**Conceptualization:** Yunhwan Kim, Ana Vivas Barber, Sunmi Lee.

**Data curation:** Yunhwan Kim, Sunmi Lee.

**Formal analysis:** Sunmi Lee.

**Methodology:** Yunhwan Kim, Ana Vivas Barber.

**Validation:** Sunmi Lee.

**Visualization:** Sunmi Lee.

**Writing – original draft:** Yunhwan Kim, Ana Vivas Barber, Sunmi Lee.

**Writing – review & editing:** Sunmi Lee.

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
