## [Decision Letter · Decision Letter 0]

21 Nov 2019

PONE-D-19-24877

Modeling influenza transmission dynamics with media coverage data of the 2009 H1N1 outbreak in Korea

PLOS ONE

Dear Dr. Lee,

Thank you for submitting your manuscript to PLOS ONE. After careful consideration, we feel that it has merit but does not fully meet PLOS ONE’s publication criteria as it currently stands. Therefore, we invite you to submit a revised version of the manuscript that addresses the points raised during the review process.

We would appreciate receiving your revised manuscript by Jan 05 2020 11:59PM. To enhance the reproducibility of your results, we recommend that if applicable you deposit your laboratory protocols in protocols.io, where a protocol can be assigned its own identifier (DOI) such that it can be cited independently in the future. For instructions see: http://journals.plos.org/plosone/s/submission-guidelines#loc-laboratory-protocols

We look forward to receiving your revised manuscript.

Kind regards,

Roberto Barrio

Academic Editor

PLOS ONE

Journal Requirements:

1. Please clarify in your Methods and Data availability statement where the data can be found, for instance in the Supplementary material or in a repository. For PLOS ONE policies on data sharing, please see https://journals.plos.org/plosone/s/data-availability. Please ensure that the data collection has been described in enough detail in the Methods section in order for another researcher to reproduce the findings.

2. please request the following from the authors and do not ping for follow up: "Please note that PLOS ONE has specific guidelines on software sharing (http://journals.plos.org/plosone/s/materials-and-software-sharing#loc-sharing-software) for manuscripts whose main purpose is the description of a new software or software package. In this case, new software must conform to the Open Source Definition (https://opensource.org/docs/osd) and be deposited in an open software archive. Please see http://journals.plos.org/plosone/s/materials-and-software-sharing#loc-depositing-software for more information on depositing your software.

Reviewers' comments:

Reviewer's Responses to Questions

**Comments to the Author**

1. Is the manuscript technically sound, and do the data support the conclusions?

Reviewer #1: Yes

Reviewer #2: Partly

2. Has the statistical analysis been performed appropriately and rigorously? 

Reviewer #1: Yes

Reviewer #2: No

3. Have the authors made all data underlying the findings in their manuscript fully available?

Reviewer #1: Yes

Reviewer #2: Yes

4. Is the manuscript presented in an intelligible fashion and written in standard English?

Reviewer #1: Yes

Reviewer #2: Yes

5. Review Comments to the Author

Reviewer #1: Please rewrite the abstract entirely. Most of what is contained in there at the moment is part of the introduction. The abstract should describe the salient points of the work you did and not the history of modeling.

Reviewer #2: Modeling influenza transmission dynamics with media coverage data

The article propose two models for the dynamics of H1N1 influenza at the human population level with different approximations for the mass media M(t). The first approximation is called the theory-based and the second one incorporates a data-based media effect term. The second approach employs data from the 2009 H1N1 incidence in South Korea as well as the real-world media coverage data in the same period.

The authors computed the basic reproduction number R0 without treatment and hospitalization. The numerical results show that the model with a theory-based media effect term and the one with a data-based media effect term produce similar outputs for a moderate value of R0.

In my opinion, the topic is interesting and useful. However, some details need to be addressed to give more clarity and robustness to the results.

• Page 2. “Xiao et al. [11] made a more realistic assumption that people are sensitive not only to the number of infected people, but also to whether the situation is getting better or worse. Thus, they built a model where the transmission rate is a function of both the number of infected people and its rate of change.” Explain better why this considered the theory-based model (page 4). Notice, that a and b are unknowns.

• Page 5. Nothing is mentioned about the equilibrium points.

• Page 5. Some comment from a dynamical point of view would help to understand why the media M(t) ( through the parameters of M(t)) do not play any role in the R0. This result seems counterintuitive.

• Line 180. The authors mentioned that the most relevant parameters are p and eta. Why? from which analysis ?

• Line 217. Please present, a little bit more of detail of the fits. How do you know that you got the global minimum in each of the 4 windows?

• Line 275. Sensitivity of final epidemic size is analyzed just based on a graph. I suggest using a metric (maybe with an integral to add cases) so results can be compared objectively.

• Line 314. “The theory-based media effect term made the result excessively dependent on the media effect, which is inconsistent with the results of previous research [6, 8].” Here is important that the authors explain why the results are inconsistent and give details.

6. PLOS authors have the option to publish the peer review history of their article (what does this mean?). If published, this will include your full peer review and any attached files.

Reviewer #1: No

Reviewer #2: No

---

## [Author Response · Author response to Decision Letter 0]

18 Jan 2020

We are grateful for the valuable comments from the two Reviewers. In this revision, we addressed all of the Reviewers’ comments and have made an effort to better explain the relevance of our results. We have also updated references and some numerical simulations as recommended by the Reviewers.

Please see the attached document of Responses to Reviewers.

---

## [Decision Letter · Decision Letter 1]

18 Feb 2020

PONE-D-19-24877R1

Modeling influenza transmission dynamics with media coverage data of the 2009 H1N1 outbreak in Korea

PLOS ONE

Dear Dr. Lee,

Thank you for submitting your manuscript to PLOS ONE. After careful consideration, we feel that it has merit but does not fully meet PLOS ONE’s publication criteria as it currently stands. Therefore, we invite you to submit a revised version of the manuscript that addresses the points raised during the review process.

We would appreciate receiving your revised manuscript by Apr 03 2020 11:59PM. To enhance the reproducibility of your results, we recommend that if applicable you deposit your laboratory protocols in protocols.io, where a protocol can be assigned its own identifier (DOI) such that it can be cited independently in the future. For instructions see: http://journals.plos.org/plosone/s/submission-guidelines#loc-laboratory-protocols

We look forward to receiving your revised manuscript.

Kind regards,

Roberto Barrio

Academic Editor

PLOS ONE

Reviewers' comments:

Reviewer's Responses to Questions

**Comments to the Author**

1. If the authors have adequately addressed your comments raised in a previous round of review and you feel that this manuscript is now acceptable for publication, you may indicate that here to bypass the “Comments to the Author” section, enter your conflict of interest statement in the “Confidential to Editor” section, and submit your "Accept" recommendation.

Reviewer #1: All comments have been addressed

Reviewer #2: (No Response)

2. Is the manuscript technically sound, and do the data support the conclusions?

Reviewer #1: Yes

Reviewer #2: Partly

3. Has the statistical analysis been performed appropriately and rigorously? 

Reviewer #1: Yes

Reviewer #2: N/A

4. Have the authors made all data underlying the findings in their manuscript fully available?

Reviewer #1: Yes

Reviewer #2: No

5. Is the manuscript presented in an intelligible fashion and written in standard English?

Reviewer #1: Yes

Reviewer #2: Yes

6. Review Comments to the Author

Reviewer #1: None. All my previous comments have been addressed. I have no other comments at the moment. My recommendation is ACCEPT

Reviewer #2: The authors improved the article taken into account my suggestions. The other reviewer just mentioned to change the abstract.

I like the idea of the article but I still have some doubts and I would like to follow the policies of PlosOne journal, in particular:

3. Experiments, statistics, and other analyses are performed to a high technical standard and are described in sufficient detail.

4. Conclusions are presented in an appropriate fashion and are supported by the data.

Regarding the effect of media varying c the authors improved the paper. I would like to see what happens with all the classes. It can be seen that with a larger value of c the peak and the total number of cases decrease. One way to do that is to show in a table and figure all the sub-populations at different times of the simulation or at least after one or two years.

Regarding the uniqueness of the solution of the inverse problem, I am not sure (as authors claim in the response) that their method guarantees a unique global solution. In the response, the authors cited a Theorem (4.5.1) by Chavent. I do not have access (and time) to the Theorem to be able to check if the conditions of the Theorem are satisfied. However, I think the authors can just extend their explanation that they gave in response letter and add the references. In that, way the authors are adding technical details (clarity) to the article as request by this journal. Notice, that the initial question is related to global uniqueness and not the local one. If authors add the paragraph to this point, they will add some strong argument to their paper.

In summary I think the authors can do these suggestions in a short time without too much work.

7. PLOS authors have the option to publish the peer review history of their article (what does this mean?). If published, this will include your full peer review and any attached files.

Reviewer #1: No

Reviewer #2: No

---

## [Author Response · Author response to Decision Letter 1]

18 Mar 2020

PONE-D-19-24877R1

Modeling influenza transmission dynamics with media coverage data of the 2009 H1N1 outbreak in Korea

PLOS ONE

We are grateful for the valuable comments from the two Reviewers. In this revision, we addressed all of the Reviewers’ comments. We have also updated references and some numerical simulations as recommended by the Reviewer. 

Reviewer #1: None. All my previous comments have been addressed. I have no other comments at the moment. My recommendation is ACCEPT

Reviewer#2: 

Regarding the effect of media varying c the authors improved the paper. I would like to see what happens with all the classes. It can be seen that with a larger value of c the peak and the total number of cases decrease. One way to do that is to show in a table and figure all the sub-populations at different times of the simulation or at least after one or two years.

Thanks for the comments. We have updated the results of other classes for Model 1 and Model 2 under three different values of media coverage (see p.8-9 and Figures 11-12 in the revised manuscript). 

Regarding the uniqueness of the solution of the inverse problem, I am not sure (as authors claim in the response) that their method guarantees a unique global solution. In the response, the authors cited a Theorem (4.5.1) by Chavent. I do not have access (and time) to the Theorem to be able to check if the conditions of the Theorem are satisfied. However, I think the authors can just extend their explanation that they gave in response letter and add the references. In that, way the authors are adding technical details (clarity) to the article as request by this journal. Notice, that the initial question is related to global uniqueness and not the local one. If authors add the paragraph to this point, they will add some strong argument to their paper.

Thanks for the comments. As the reviewer suggested, we have addressed this issue in the revised manuscript (see p. 7-8). The references below have been included in the revised manuscript as well. 

R. Aster, B. Borchers, and C. Thurber, editors. Parameter Estimation and Inverse Problems. Elsevier, second edition, 2012. ISBN 978-0-128-10092-9. 

F. Yaman, V. G. Yakhno, and R. Potthast. A Survey on Inverse Problems for Applied Sciences. Math. Probl. Eng., 2013, 2013. doi: 10.1155/2013/ 976837. 

G. Chavent. Nonlinear Least Squares for Inverse Problems: Theoretical Foundations and Step-by-Step Guide for Applications. Springer, 2010. doi: 10.1007/978-90-481-2785-6. 

A. R. Conn, N. I.M. Gould, and P. L. Toint, editors. Trust Region Methods. SIAM, 2000. 

Eber Dantas, Michel Tosin, Americo Cunha Jr∗ Calibration of a SEIR–SEI epidemic model to describe the Zika virus outbreak in Applied Mathematics and Computation. 338 (2018) 249–259.

---

## [Decision Letter · Decision Letter 2]

14 Apr 2020

PONE-D-19-24877R2

Modeling influenza transmission dynamics with media coverage data of the 2009 H1N1 outbreak in Korea

PLOS ONE

Dear Dr. Lee,

Thank you for submitting your manuscript to PLOS ONE. After careful consideration, we feel that it has merit but does not fully meet PLOS ONE’s publication criteria as it currently stands. Therefore, we invite you to submit a revised version of the manuscript that addresses the points raised during the review process.

We would appreciate receiving your revised manuscript by May 29 2020 11:59PM. To enhance the reproducibility of your results, we recommend that if applicable you deposit your laboratory protocols in protocols.io, where a protocol can be assigned its own identifier (DOI) such that it can be cited independently in the future. For instructions see: http://journals.plos.org/plosone/s/submission-guidelines#loc-laboratory-protocols

We look forward to receiving your revised manuscript.

Kind regards,

Roberto Barrio

Academic Editor

PLOS ONE

Reviewers' comments:

Reviewer's Responses to Questions

**Comments to the Author**

1. If the authors have adequately addressed your comments raised in a previous round of review and you feel that this manuscript is now acceptable for publication, you may indicate that here to bypass the “Comments to the Author” section, enter your conflict of interest statement in the “Confidential to Editor” section, and submit your "Accept" recommendation.

Reviewer #2: All comments have been addressed

2. Is the manuscript technically sound, and do the data support the conclusions?

Reviewer #2: Yes

3. Has the statistical analysis been performed appropriately and rigorously? 

Reviewer #2: Yes

4. Have the authors made all data underlying the findings in their manuscript fully available?

Reviewer #2: Yes

5. Is the manuscript presented in an intelligible fashion and written in standard English?

Reviewer #2: Yes

6. Review Comments to the Author

Reviewer #2: The authors improved the paper. As I mentioned initially the article is interesting and useful. I like the idea. In the previous review, I suggested to add a figure with all the populations. The authors added Figs 11-12, but without the Susceptible and Vaccinated classes. This is a small thing to add.

I like the article and the proposed the idea, and I think after that the article is ready to be accepted, unless the figures show some unexpected result.

7. PLOS authors have the option to publish the peer review history of their article (what does this mean?). If published, this will include your full peer review and any attached files.

Reviewer #2: No

---

## [Author Response · Author response to Decision Letter 2]

14 Apr 2020

PONE-D-19-24877R2

Modeling influenza transmission dynamics with media coverage data of the 2009 H1N1 outbreak in Korea

PLOS ONE

We are grateful for the valuable comments from the Reviewer. In this revision, we addressed all of the Reviewers’ comments. We have updated numerical simulations as recommended by the Reviewer. 

Reviewer #2: The authors improved the paper. As I mentioned initially the article is interesting and useful. I like the idea. In the previous review, I suggested to add a figure with all the populations. The authors added Figs 11-12, but without the Susceptible and Vaccinated classes. This is a small thing to add.

Thanks for the comments. We have updated the results of all classes for Model 1 and Model 2 under three different values of media coverage (see pp.20-21 Figures 11-12 in the revised manuscript).

---

## [Decision Letter · Decision Letter 3]

20 Apr 2020

Modeling influenza transmission dynamics with media coverage data of the 2009 H1N1 outbreak in Korea

PONE-D-19-24877R3

Dear Dr. Lee,

We are pleased to inform you that your manuscript has been judged scientifically suitable for publication and will be formally accepted for publication once it complies with all outstanding technical requirements.

With kind regards,

Roberto Barrio

Academic Editor

PLOS ONE

Reviewer's Responses to Questions

**Comments to the Author**

1. If the authors have adequately addressed your comments raised in a previous round of review and you feel that this manuscript is now acceptable for publication, you may indicate that here to bypass the “Comments to the Author” section, enter your conflict of interest statement in the “Confidential to Editor” section, and submit your "Accept" recommendation.

Reviewer #2: All comments have been addressed

2. Is the manuscript technically sound, and do the data support the conclusions?

Reviewer #2: Yes

3. Has the statistical analysis been performed appropriately and rigorously? 

Reviewer #2: Yes

4. Have the authors made all data underlying the findings in their manuscript fully available?

Reviewer #2: Yes

5. Is the manuscript presented in an intelligible fashion and written in standard English?

Reviewer #2: Yes

6. Review Comments to the Author

Reviewer #2: Questions were addressed, and is ready to be published. I think the article is interesting and useful. Some small doubts about the uniqueness of the global problem since there are four time intervals.

7. PLOS authors have the option to publish the peer review history of their article (what does this mean?). If published, this will include your full peer review and any attached files.

Reviewer #2: No

---

## [Editor Report · Acceptance letter]

6 May 2020

PONE-D-19-24877R3 

Modeling influenza transmission dynamics with media coverage data of the 2009 H1N1 outbreak in Korea 

Dear Dr. Lee:

I am pleased to inform you that your manuscript has been deemed suitable for publication in PLOS ONE. Congratulations! Your manuscript is now with our production department. 

With kind regards,

on behalf of

Dr. Roberto Barrio 

Academic Editor

PLOS ONE